# Non-linear and Interaction Analyses of Biomarkers for Organ Dysfunctions as Predictive Markers for Sepsis: A Nationwide Retrospective Study

**DOI:** 10.3390/jpm12010044

**Published:** 2022-01-04

**Authors:** Yutaka Umemura, Kazuma Yamakawa, Shuhei Murao, Yumi Mitsuyama, Hiroshi Ogura, Satoshi Fujimi

**Affiliations:** 1Osaka General Medical Center, Division of Trauma and Surgical Critical Care, Osaka 558-5885, Japan; shmu20268271@gmail.com (S.M.); goto8940@gmail.com (Y.M.); sfujimi40@nifty.com (S.F.); 2Department of Traumatology and Acute Critical Medicine, Graduate School of Medicine, Osaka University, Osaka 565-0871, Japan; ogura@hp-emerg.med.osaka-u.ac.jp; 3Department of Emergency Medicine, Osaka Medical and Pharmaceutical University, Takatsuki 569-8686, Japan; kazuma.yamakawa@ompu.ac.jp

**Keywords:** sepsis, biomarkers, mortality, intensive care units, prognosis, organ dysfunction

## Abstract

The Sequential Organ Failure Assessment (SOFA) score is predominantly used to assess the severity of organ dysfunction in sepsis. However, differences in prognostic value between SOFA subscores have not been sufficiently evaluated. This retrospective observational study used a large-scale database containing about 30 million patients. Among them, we included 38,869 adult patients with sepsis from 2006 to 2019. The cardiovascular and neurological subscores were calculated by a modified method. Associations between the biomarkers of the SOFA components and mortality were examined using restricted cubic spline analyses, which showed that an increase in the total modified SOFA score was linearly associated with increased mortality. However, the prognostic association of subscores varied widely: platelet count showed a J-shaped association, creatinine showed an inverted J-shaped association, and bilirubin showed only a weak association. We also evaluated interaction effects on mortality between an increase of one subscore and another. The joint odds ratios on mortality of two modified SOFA subscores were synergistically increased compared to the sum of the single odds ratios, especially in cardiovascular-neurological, coagulation-hepatic, and renal-hepatic combinations. In conclusion, total modified SOFA score was associated with increased mortality despite the varied prognostic associations of the subscores, possibly because interactions between subscores synergistically enhanced prognostic accuracy.

## 1. Introduction

Sepsis is defined as life-threatening organ dysfunction caused by an overwhelming host response to an infection [1,2]. Despite the progress in its medical management over the past few decades, sepsis remains an important global health problem, with approximately 11 million sepsis-related deaths reported in 2017 [3].

Development of multiple organ dysfunction is the most important clinical feature during sepsis. The predominant index used to assess organ dysfunction in sepsis is the Sequential Organ Failure Assessment (SOFA) score [4]. A number of studies have provided definite evidence indicating that an increase of the SOFA score in sepsis is associated with an increased probability of mortality [5,6,7].

However, the clinical significance of the SOFA subscores has not been sufficiently evaluated so far. For example, weighting in each component of SOFA, such as platelet count, creatinine level, and bilirubin level, was defined to be homogeneous, whereas the pathophysiology of organ dysfunctions is naturally heterogeneous. In addition, the cutoff point of each SOFA subscore has been determined based on the consensus of experts and has not been sufficiently evaluated statistically regarding its association with outcomes. Furthermore, there is little evidence on the impact on mortality when multiple components in the SOFA subscores are simultaneously increased. The pathophysiology of multiple organ dysfunction cannot simply be considered as an additive combination of each single organ dysfunction. Dysfunctional organs may impact other remote organs through complex, and incompletely understood, biological communication processes. This cross-interaction is so-called organ crosstalk and is considered to accelerate organ dysfunctions [8]. In other words, certain combinations of SOFA subscores might lead to a synergistically increased mortality in sepsis.

Therefore, detailed prognostic associations and interaction effects of the SOFA subscores need to be elucidated to verify the clinical significance of the SOFA score. To date, some evidence has supported the hypothesis that some biomarkers would non-linearly associate with clinical outcomes [9,10]. However, the current clinical evidence regarding the non-linear relation between the components of the SOFA subscores and patient prognosis remains limited. Similarly, there is little evidence of the interaction prognostic effects caused by combinations of the SOFA subscores.

The present study thus aimed to evaluate 1) the detailed, non-linear association between SOFA subscores and mortality and 2) the interaction effects of a simultaneous increase in SOFA subscores on mortality in sepsis using a large, nationwide registry database.

## 2. Materials and Methods

### 2.1. Design and Setting

This retrospective observational study used a part of a large-scale database covering approximately 23% of acute-care hospitals in Japan that contained about 30 million patients as of October 2019 (Medical Data Vision Co., Tokyo, Japan). The database includes data on age, sex, primary diagnoses, concomitant diagnoses, complication diagnoses, procedures, prescriptions, discharge status, and laboratory tests. In this database, the diagnoses are recorded using International Classification of Diseases Tenth Revision (ICD-10) codes. Among the overall patient data registered in the database, this study included patient data from 42 acute-care hospitals having laboratory data.

This study followed the principles of the Declaration of Helsinki, and the study protocol was approved by the Institutional Review Board for Clinical Research of Osaka General Medical Center (IRB No. S201916015, 21 April 2021). The board waived the requirement for informed consent because of the anonymous nature of the data and because no information on individual patients, hospitals, or treating physicians was obtained.

### 2.2. Participants

We included all patients who required unplanned hospital admission and were diagnosed as having sepsis from 1 February 2006 to 31 December 2019. In this study, sepsis was defined as having a proven/suspected infection and the development of organ dysfunction according to the Sepsis-3 criteria. Organ dysfunction in this study was defined as a total SOFA score of ≥2 points at the time of admission because pre-existing organ dysfunction was not sufficiently recorded and Sepsis-3 suggested that the baseline SOFA score could be assumed to be zero in patients not known to have preexisting organ dysfunction [1]. A proven/suspected infection was defined as having any of the infection-related ICD-10 codes previously proposed by the Institute for Health Metrics and Evaluation [3] in the primary diagnosis or the diagnosis that triggered hospitalization. The detailed set of the ICD-10 codes for the presence of infection is listed in Appendix A. We excluded patients who were admitted to hospital with a diagnosis of sepsis more than once during the study period (i.e., had a second or subsequent record of admission).

### 2.3. Data Collection

We collected the following data on baseline patient characteristics: age, sex, height, weight, Charlson Comorbidity Index [11], intensive care unit admission, anatomical site of infection, organ dysfunction, use of catecholamines, and laboratory tests including platelet count, white blood cell count, bilirubin, creatinine, prothrombin time, C-reactive protein, blood glucose, lactate dehydrogenase, total protein, albumin, uric acid, blood urea nitrogen, and creatine kinase.

We used previously published ICD-10 coding algorithms for defining Charlson comorbidities [12]. Organ dysfunction was evaluated by ICD-10 coding algorithms previously published by Angus et al. [13] and Martin et al. [14]. The presence and severity of organ dysfunction at the time of hospital admission were also evaluated according to the SOFA score.

The renal, hepatic, and coagulation subscores of SOFA were calculated based on laboratory tests for creatinine, bilirubin, and platelet count. The cardiovascular and neurological subscores were calculated by a previously published modified method [15] (Appendix A) because mean arterial pressure and Glasgow Coma Scale scores were not recorded in this study registry. The respiratory score was also calculated by modified methods, but it was not included in the statistical analyses because the modified method regarding the respiratory component widely differed from the original one due to the lack of data on SpO_2_, arterial blood gas values, and mechanical ventilator settings. The Japan Coma Scale, which was used for calculating the neurological subscore instead of the Glasgow Coma Scale, has four main grades (grade 0: alert; grade 1: possible verbal response without any stimulation, not lucid; grade 2: possible eye-opening, verbal and motor response upon stimulation; and grade 3: no eye-opening and coma upon stimulation). The primary outcome measure was all-cause in-hospital mortality.

### 2.4. Statistical Analysis

Descriptive statistics were calculated as group medians with the first and third quartiles for continuous variables and as frequencies with percentages for categorical variables. The differences in patient characteristics, general laboratory test results, and severity of illness scores between survivors and non-survivors were evaluated by the Mann-Whitney U test or chi-square test.

The detailed association between the biomarkers of the SOFA subscores and mortality was evaluated using non-linear methods, and the interaction effects of modified SOFA subscores on mortality were evaluated by dichotomizing each SOFA subscore into the higher score (2-point or more) or the lower score (1-point or less). To evaluate the non-linear association between mortality and the biomarkers and to visually provide comprehensible information about the significances of each cutoff point of the SOFA subscores, we constructed restricted cubic spline curves using logistic regression models. The knot values were determined based on Harrell’s recommended percentiles, with the knots placed at equally spaced percentiles of the original variable’s marginal distribution [16]. The number of knots in each analysis was determined by the Wald test in such way that the explanatory variables at all sections divided by the knots were significant (Appendix A) [17].

To assess the additive interactions in mortality according to the simultaneous increase in every two SOFA subscores, we evaluated whether the joint effects of two subscores with a positive effect on mortality (2-point or more increase) were higher compared to the sum of the single effects of the subscores with a positive effect. To capture the proportion of the joint effect attributable to additive interaction, we calculated relative excess risks due to interaction (RERI) and the attributable proportion [18]. Statistical significances for interaction were evaluated by logistic regression analyses including two-way interaction terms between the subscores with a positive effect. Univariate and multivariable logistic regression analyses were also conducted to evaluate the risk of death in accordance with every 1-point increase in the SOFA subscores. There were possible confounding effects on mortality between the SOFA subscores: in other words, patients having one organ dysfunction tended to have additional ones, which might enhance the risk of death. We therefore fitted multivariable analyses for SOFA subscore on mortality, including the other four SOFA subscores as covariates, to adjust for possible confounding effects and evaluate the independent risk of death associated with an increase of the subscore. We also conducted receiver operating characteristic (ROC) analyses to evaluate the predictive value of the SOFA score for mortality.

As there were moderate proportions of missing data in the biomarkers (2548 (6.6%) for creatinine, 3933 (10.1%) for bilirubin, and 2454 (6.3%) for platelet count), missing values were not imputed in any of the regression models because the cause of the missing data for the biomarkers could not be assumed to be “missing at random”.

To evaluate the possible differences of findings according to the time-related changes in the management of sepsis and updates of the Surviving Sepsis guidelines, we performed subgroup analysis based on the time of hospital admission (the earlier group: 1 February 2006 to 31 December 2012 and the recent group: 1 January 2013 to 31 December 2019).

All hypotheses were two-sided, and a *p* value of <0.05 was considered to indicate statistical significance. Because of the underpowered nature of the interaction analysis, we used a two-sided significance level of 20% with statistical inferences for the analyses of interaction [19]. All statistical analyses were conducted using STATA Data Analysis and Statistical Software version 15.0 (StataCorp LLC, College Station, TX, USA).

## 3. Results

### 3.1. Study Population

During the study period, 186,125 patients were diagnosed as having an infectious disease and required unplanned hospital admission. Among them, we included 41,027 patients who had a total SOFA score of 2 points or more at the time of admission. After excluding 2158 patients due to a second or subsequent admission to hospital with a diagnosis of sepsis, we included 38,869 patients in the final study cohort. Among the study patients, 34,088 patients survived to discharge and 4781 patients died (Figure 1).

For the subgroup analysis, we classified the study population based on the time of hospital admission. Patients who were admitted to hospital from 1 February 2006 to 31 December 2012 were classified into the earlier group (*n* = 19,201), and patients who were admitted from 1 January 2013 to 31 December 2019 were classified into the recent group (*n* = 19,699).

Baseline characteristics, general laboratory test data, and modified SOFA score in the survivors and non-survivors are shown in Table 1. Although the non-survivors were significantly older than the survivors (84 vs. 79 years old, *p* < 0.001), the sex ratio and Charlson Comorbidity Index were not different between the groups. The distribution of the anatomical site of infection varied significantly between the groups (*p* < 0.001). The levels of the most recorded laboratory tests, except for that of white blood cell count, were significantly different between groups. Similarly, the total modified SOFA score was significantly higher in the non-survivors (3 vs. 4, median, *p* < 0.001).

### 3.2. Non-linear Associations between Mortality and Biomarkers

Non-linear associations between in-hospital mortality and the components of the SOFA score are described as restricted cubic spline curves in Figure 2. An increase in total SOFA score was sharply and almost linearly associated with increased mortality. When the cutoff value was defined as 5 points, the sensitivity, specificity, and area under the ROC curve of the SOFA score for mortality were 38.1%, 86.3%, and 0.676, respectively. Although there was little change in predicted mortality as indicated by platelet counts within a range of 150,000 to 500,000/µL, the risk of death rose sharply as the platelet counts decreased below a level of approximately 150,000/µL. The risk of mortality rose sharply as the levels of creatinine increased from normal range to approximately 3.5 mg/dL, but there was no remarkable increase in mortality according to the increase of creatinine levels when the levels were higher than 5.0 mg/dL. Similarly, an increase in mortality in accordance with the level of bilirubin was observed only when the levels were below approximately 2.0 mg/dL and was not remarkable when the levels increased above 2.0 mg/dL.

The results of restricted cubic spline analyses in the two subgroups based on the time of hospital admission are shown in Appendix A. The associations between the biomarkers and mortality were similar between the earlier and recent subgroups.

### 3.3. Risk of Death according to the Increase of SOFA Subscores

The odds ratio for in-hospital death for each point of the SOFA subscores compared to its 0 point is shown in Figure 3. Increases in the neurological and coagulation subscores were associated with a significant and linear increase in the risk of death. The cardiovascular subscore was also significantly associated with an increase in the risk of death. An increase in the renal subscore was also associated with a significant increase in the risk of death within a range of 1 to 3 points. However, the odds ratio for the subscore of 4 points was relatively lower compared to that for the other subscore points. Similarly, the mortality odds ratio for a hepatic subscore of 4 points was significant but lower compared to those for 1 to 3 points.

We also evaluated the risk of death in accordance with every 1-point increase in the SOFA subscores by univariate and multivariable logistic regression analyses (Appendix A). Multivariable logistic regression analysis revealed the neurological, cardiovascular, coagulation, and renal scores to be independently associated with a significant increase in mortality, whereas the hepatic score was not.

### 3.4. Interaction between SOFA Subscores

The cross-interaction effects on mortality between SOFA subscores are summarized as a heat map corresponding to the attributable proportion of the joint odds ratios of the SOFA subscores positive (2-point or more increase) on mortality (Figure 4). Each cell contains the following: on the first line, the joint mortality odds ratios of the simultaneous increase in two SOFA subscores above 2 points; on the second line, the attributable proportion of the joint effects to interaction between the increases of the two SOFA subscores; and on the third line, the *p* value for the two-way interactions on mortality between the increases of the two SOFA subscores. As a result, the joint odds ratios on mortality of two SOFA scores were synergistically increased compared to the sum of the single odds ratios with high attributable proportions, and the interactions were statistically significant in almost all subgroups. In particular, the cardiovascular-neurological, coagulation-hepatic, and renal-hepatic combinations were associated with remarkably higher attributable proportions (77.4%, 78.3%, and 99.9%, respectively).

## 4. Discussion

### 4.1. Prognostic Association in SOFA Subscores

The present study showed that an increase in total SOFA score was almost linearly associated with increased mortality, whereas all three biomarkers of the SOFA subscores were non-linearly associated with the risk of death. The high prognostic value of the total SOFA score has been definitely proven by multiple lines of evidence [20]. Similarly, the three biomarkers of the SOFA subscores have long been evaluated as biomarkers predictive of mortality [21,22,23]. However, there is little evidence regarding the non-linear prognostic associations of the SOFA subscores, partly because an extremely large sample size is required to construct a non-linear prediction line with high confidence. The methods and results of the current study were presented according to the STROBE Statement checklist (Appendix A).

Blood coagulation disorders are invariably present in patients with sepsis and play a critical role in the progression of multiple organ dysfunction syndrome. In this study, platelet count showed a J-shaped association with mortality, namely, it was associated with mortality only when it decreased below a level of approximately 150,000/µL. This finding agreed with that in a previous study evaluating hemostatic biomarkers using multicenter cohort data [10] and supported the position of the current cutoff point in the SOFA coagulation subscore. Actually, an increase in the coagulation subscore was consistently associated with a significant increase in mortality.

Serum creatinine remains the gold standard for acute kidney injury (AKI), which has long been recognized as an important complication of sepsis independently associated with mortality [23,24,25]. In the present study, creatinine showed an inverted J-shaped association, namely, it was associated with mortality when increased from the normal range to approximately 3.5 mg/dL. However, the association with increased mortality was lost when the value increased to above 5.0 mg/dL. The level of creatinine in sepsis-induced AKI at the time of diagnosis does not generally increase above 5.0 mg/dL but typically ranges between 1.0 to 3.0 mg/dL [26,27]. Actually, both the Risk, Injury, Failure, Loss, and End-stage renal failure (RIFLE) and Kidney Disease Improving Global Outcomes (KDIGO) criteria for AKI set the creatinine thresholds for worst classification of renal dysfunction at 4.0 mg/dL [28,29]. Therefore, the reason for the relatively lower risk in patients with extremely high creatinine levels might be attributable to the increase in the creatinine level above 5.0 mg/dL possibly not reflecting the acute progression of renal dysfunction by sepsis but rather reflecting the presence of chronic kidney disease.

Similarly, the total bilirubin level in the acute phase of sepsis other than in the presence of biliary tract infection has been reported to be associated with increased mortality, but typically, it ranges within a normal to slightly higher level [22,30]. Extremely high levels of bilirubin (e.g., above 6.0 mg/dL) would mainly reflect the presence of biliary tract infection or chronic liver failure, and thus they were not associated with a remarkable increase in the risk of death in the present study when including all anatomical sites of infection.

### 4.2. Interaction of Subscores

Despite the heterogeneous and partly suboptimal prognostic values of the subscores, the total SOFA score was sharply and linearly associated with the risk of death. One possible explanation for this high prognostic value of the total SOFA score may be attributable to the interactions of organ dysfunctions, so-called organ crosstalk [8]. For example, the prognostic value of bilirubin (hepatic subscore) was found not to be so high in the present study, but its prognostic association in the combination of hepatic and other subscores was synergistically increased. These interactions on mortality would reflect well-known pathophysiologies in organ crosstalk, such as hepatorenal syndrome or hepatic encephalopathy [31,32].

In addition, overwhelming activations of the inflammation and coagulation systems are essential responses for host protection against microbial invasion, but they can also facilitate multiple organ damage through mutual interactions leading to disseminated intravascular coagulation [33].

Along with these pathophysiological mechanisms, therapeutic reasons might possibly contribute to the interactions of some organ dysfunctions evaluated by SOFA. For example, patients with severe circulatory disorders are typically anesthetized and intubated and thus tend to have higher respiratory and neurological subscores. These interactions would therefore synergistically enhance the prognostic value of the total SOFA score, even though the prognostic value and cutoff points of some components would be suboptimal as single biomarkers.

In summary, the present study revealed differences in the weight of the SOFA sub-components. The weight of the SOFA sub-components has so far been artificially defined to be homogeneous, whereas the pathophysiology of organ dysfunctions is naturally heterogeneous and they complicatedly affect one another. Further, our study clearly provided clinical evidence of organ cross-talk: in other words, the pathophysiology of multiple organ dysfunction is not simply one of additive combinations of each single organ dysfunction but one of a synergistically worsened condition with an extremely higher risk of death. It is therefore important when evaluating the severity of organ dysfunctions with the SOFA score to consider which organ and which combinations of organs are mainly responsible for the elevation of the total score. Our findings may also be expected to be the foundation of future studies to construct a new advanced method to evaluate the severity of organ dysfunction.

### 4.3. Limitations

First, the diagnoses recorded in an administrative claims databases are generally less accurate than those in planned prospective studies because misclassification, underestimation, or overestimation of diagnoses might have occurred. Second, we used a modified SOFA score in the cardiovascular, respiratory, and neurological components because P/F ratio, vital signs, and the Glasgow Coma Scale score were not recorded in this study registry. In particular, the modified method regarding the respiratory component widely differed from the original one, and thus we decided not to include this component in the statistical analyses. Third, this study could not distinguish between pre-existing and newly developed organ dysfunctions because the baseline SOFA score is not recorded in this study registry. Fourth, this study included anesthetized and intubated patients, who tend to have higher neurological and cardiovascular subscores compared to other subscores. Fifth, the heterogenous pathophysiology of sepsis based on differences in the anatomical site of infection might have also affected the study findings. To resolve these potential imbalances, our results should be validated in another large-scale clinical study.

## 5. Conclusions

Despite the widely varied prognostic associations of SOFA subscores, the total SOFA score was sharply and almost linearly associated with increased mortality. Cross-interactions between subscores synergistically enhanced its prognostic associations and might be responsible for the high prognostic accuracy of the total SOFA score.

## Figures and Tables

**Figure 1 jpm-12-00044-f001:**
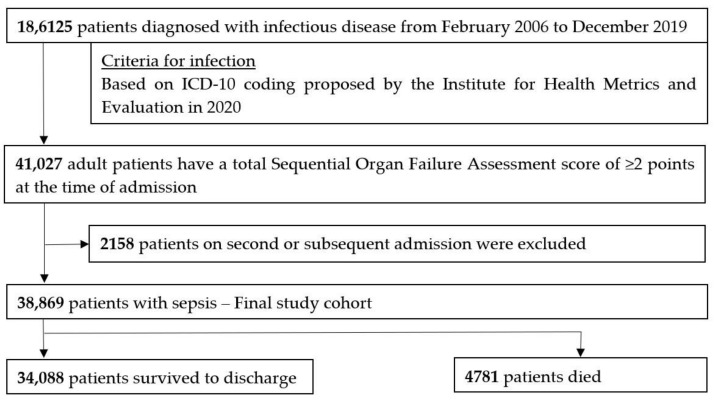
Patient flow diagram. ICD-10: International Classification of Diseases Tenth Revision.

**Figure 2 jpm-12-00044-f002:**
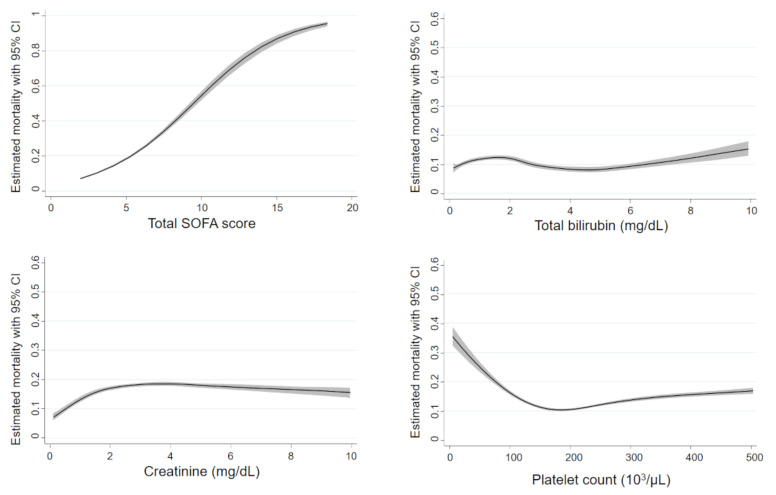
Non-linear association between mortality and the components of the SOFA. The black line represents the fitted line of the association between biomarkers and the estimated mortality risk, and the shaded region represents the 95% confidence interval. Total SOFA score was the sum of the five SOFA subscores except for the respiratory subscore. SOFA: Sequential Organ Failure Assessment, CI: Confidence Interval.

**Figure 3 jpm-12-00044-f003:**
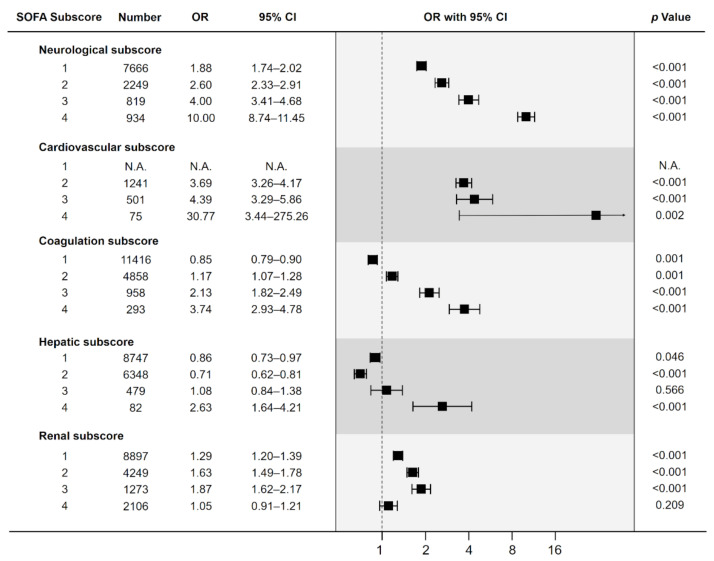
Mortality odds ratios according to the increase in SOFA subscores, compared to 0 points for each subscore. CI: Confidence Interval, OR: Odds Ratio, SOFA: Sequential Organ Failure Assessment, N.A.: Not Applicable.

**Figure 4 jpm-12-00044-f004:**
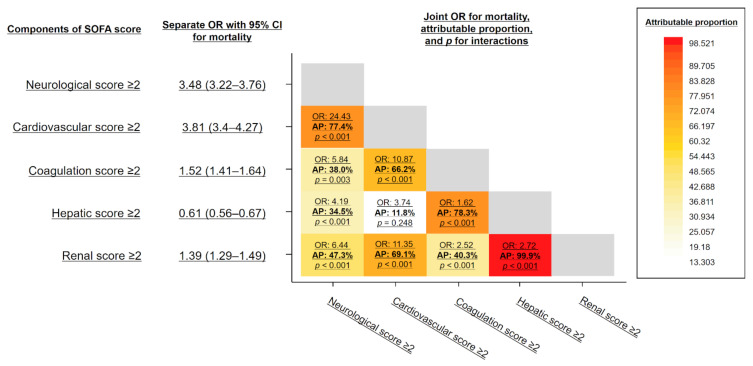
Heat map corresponding to the attributable proportion of the joint effects to interaction between the increases of two SOFA subscores. Each cell contains the following: (1) on the first line: the joint mortality odds ratios of the simultaneous increase in the two SOFA subscores above 2 points, (2) on the second line: the attributable proportion of the joint effects to interaction between the increases of the two SOFA subscores; and (3) on the third line: the *p* value for the two-way interactions on mortality between the increase of the two SOFA subscores. The color key histogram shows the attributable proportion to interaction within each color bar. CI: Confidence Interval, OR: Odds Ratio, SOFA: Sequential Organ Failure Assessment, AP: Attributable Proportion.

**Table 1 jpm-12-00044-t001:** Patient characteristics.

Patient Characteristics	Total	Survivors	Non-Survivors	*p* Value
	*n* = 38,869	*n* = 34,088	*n* = 4781	
Age, years	80 (69–86)	79 (68–86)	84 (76–89)	<0.001
Sex, male	22,904 (58.9%)	20,132 (59.1%)	2772 (58%)	0.155
Height (cm)	158 (150–165)	158 (150–165)	155 (148–163)	<0.001
Weight (kg)	53.5 (45–63)	54.1 (45.8–63.6)	47.8 (40–56.2)	<0.001
Charlson Comorbidity Index	5 (2–9)	5 (2–10)	5 (2–9)	<0.001
Anatomical site of infection				<0.001
Respiratory	16,608 (42.7%)	13,640 (40%)	2968 (62.1%)	
Abdominal	8777 (22.6%)	7965 (23.4%)	812 (17%)	
Urinary tract	6477 (16.7%)	6046 (17.7%)	431 (9%)	
Bone/soft tissue	1754 (4.5%)	1645 (4.8%)	109 (2.3%)	
Central nervous system	555 (1.4%)	496 (1.5%)	59 (1.2%)	
Cardiovascular	420 (1.1%)	376 (1.1%)	44 (0.9%)	
Other/Unclassifiable	4278 (11%)	3920 (11.5%)	358 (7.5%)	
White blood cell count (10^3^/µL)	99 (68–138.4)	99 (68.5–137.7)	100 (66–145)	0.200
C-reactive protein (mg/dL)	8.2 (2.8–16.1)	7.9 (2.6–15.7)	10.2 (4.4–18.3)	<0.001
Platelet count (10^3^/µL)	15.8 (11.8–21.9)	15.8 (12–21.8)	15.9 (10.7–23.2)	<0.001
Bilirubin (mg/dL)	1 (0.6–1.6)	1 (0.6–1.7)	0.8 (0.5–1.39)	<0.001
Creatinine (mg/dL)	1.06 (0.74–1.69)	1.04 (0.74–1.64)	1.21 (0.77–2)	<0.001
Prothrombin time (%)	76.3 (62–89)	77.5 (64–90)	67.8 (51.4–82.2)	<0.001
Glucose (mg/dL)	130 (108–165)	129 (109–163)	133 (104–178)	0.118
Albumin (g/dL)	3.3 (2.8–3.8)	3.4 (2.9–3.8)	2.8 (2.3–3.3)	<0.001
Blood urea nitrogen (mg/dL)	23 (15.8–36.1)	22 (15.2–34.1)	32.4 (21.1–50)	<0.001
Modified SOFA score total	3 (2–4)	3 (2–4)	4 (3–6)	<0.001

SOFA: Sequential Organ Failure Assessment.

## Data Availability

The statistical codes and full dataset are available from the corresponding author.

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
