# Peer review of "Non-linear and Interaction Analyses of Biomarkers for Organ Dysfunctions as Predictive Markers for Sepsis: A Nationwide Retrospective Study"

_jpm, 2022, doi:10.3390/jpm12010044_

Round 1
Reviewer 1 Report
**General comments**
The authors evaluated the association between biomarker components of the SOFA score and two scores indicating dysfunctions of the cardiovascular and neurological system and mortality using a model that allowed for nonlinear relationships as well as interactions between variables representing organ dysfunction. I congratulate the authors on their ambitious attempt to refine the scoring of organ dysfunction severity in sepsis. I was pleased to notice advanced methods such as splines and multiple imputation of missing data, though not all aspects of the presentation of these analyses are clear to me. I think the manuscript can be further improved by the authors in line with comments listed below.
**Major issues**
1. What is the take home message for clinicians? There is some ambiguity in the discussion: on the one hand, the authors state that there are both nonlinearities and interactions between organ systems. On the other, they rationalise the use of a single composite score with equal weighting of particular organ dysfunctions while disregarding the cross-talk between different systems. Admittedly playing the devil’s advocate here, I will ask the question: what’s the incentive for future exploration of these intricacies if the modified SOFA score seems to work just fine?
2. The title and abstract are misleading: the degree of deviation from the original SOFA score in my opinion warrants more recognition in these sections. One could argue that „biomarker components of the SOFA score” is a more accurate indication of this manuscript’s focus. The fact that patients with pre-existing organ dysfuntion could be treated as de-novo diagnosis of sepsis should be highlighted in the methods section.
3. Did the authors follow any reporting guidelines? I could not find exact number of missing observations in the text or supplement. Stating that the amount of missingness was moderate is definitely not enough and details of the multiple imputation process need to be described - in the current form I would not know how to reproduce these findings and how successful the imputation actually was. Furthermore, I would argue that the MAR assumption is shaky regarding e.g. bilirubin - one could argue that bilirubin was likely to be normal in cases where the attending physician did not see indications to measure it.
4. Synergistic effect were analysed using a dichotomy (≥2 points in a particular subscore), while the manuscript overall seems to argue for a refined non-linear appraoch. How are these two approaches compatible?
**Minor issues**
1. The timeframe is extremely wide - this can be viewed as both a strength and limitation. As the definition of sepsis (and treatment guidelines) changed several times over the years, a sensitivity analysis looking more narrowly at specific periods of time could provide additional perspective on the heterogeneity of studied relationships.
2. From my arguably limited understanding, Prof. Harrell recommends 5 knots in large datasets - what is the advantage of tweaking the knot positions based on statistical significance? I assume such supervised learning distorts the properties of p-values and confidence intervals. What were the final knot positions (percentile of data distribution)?
3. Please use the term multivariable not multivariate in this context (there is one dependent variable).
4. The term confounding makes sense in a causal inference framework - is this the author’s intention? If so, STROBE guidelines and the exposure-outcome framework should be applied consistently. If not and prediction is of interest, the TRIPOD guidelines apply.
5. What treatment effects to the authors have in mind in the last paragraph of the methods section?
6. What does it mean that scores are „dependently associated” with an increase in mortality?
7. I would avoid the use of the word „skyrocketed” and ask the authors to look closely at confidence interval around the cardiovascular subscore of 4 - it’s just not enough data to draw conclusions about this group of patients.
8. „Despite the heterogenetic (and partly suboptimal) prognostic values” - did you mean heterogeneous? I would advice to use parentheses as sparsely as possible and rephrase the sentence.
9. The authors argue that patients who are anesthesized and intubated automatically have higher scores in other systems. However, GCS is often assumed or carried forward using the last value before the patients received medications that alter mental status. Furthermore, SOFA uses the PaO2/FiO2 ratio, so intubation itself is less relevant than pulmonary function.
10. In the limitations the authors state that detailed amounts of catecholamines were not registered. How was the cardiovascular score calculated then?
**Other issues**
I had to re-read some sections, particularly methods - several sentences feel like they could have been made more readable but I will not make harsh judgements as I am not a native speaker myself. There are several typos in the text.
Author Response
We would like to thank all of the editors and reviewers for helping us improve our article. We revised our manuscript according to the comments and suggestions given by the reviewers. We also highlighted additional changes in the revised manuscript in red color with underline. Below are our point-by-point responses to the reviewers’ comments.
Reviewer #1:
**General comments**
The authors evaluated the association between biomarker components of the SOFA score and two scores indicating dysfunctions of the cardiovascular and neurological system and mortality using a model that allowed for nonlinear relationships as well as interactions between variables representing organ dysfunction. I congratulate the authors on their ambitious attempt to refine the scoring of organ dysfuncton severity in sepsis. I was pleased to notice advanced methods such as splines and multiple imputation of missing data, though not all aspects of the presentation of these analyses are clear to me. I think the manuscript can be further improved by the authors in line with comments listed below.
**Major issues**
- What is the take home message for clinicians? There is some ambiguity in the discussion: on the one hand, the authors state that there are both nonlinearities and interactions between organ systems. On the other, they rationalise the use of a single composite score with equal weighting of particular organ dysfunctions while disregarding the cross-talk between different systems. Admittedly playing the devil’s advocate here, I will ask the question: what’s the incentive for future exploration of these intricacies if the modified SOFA score seems to work just fine?
Reply: Thank you for your valuable comments and question. As you pointed out, the current version of our manuscript may not sufficiently address the clinical significance and importance of the study findings. Organ dysfunction, predominantly evaluated by the SOFA score, is a key pathophysiology in sepsis. The current study provides clinically important insights into the pathophysiology of organ dysfunction in sepsis regarding the following points:
- This study revealed the differences in the weight of SOFA sub-components, which are artificially defined to be homogeneous, whereas the pathophysiology of organ dysfunctions is naturally heterogeneous.
- This study clinically demonstrated that the pathophysiology of multiple organ dysfunction was not simply one of additive combinations of each single organ dysfunction but one of a synergistically worsened condition with an extremely higher risk of death.
Based on these insights, it is therefore important when evaluating the severity of organ dysfunctions with the SOFA score to consider which organs and which combinations of organs are mainly responsible for the elevation of total score. To address these points more clearly, we have added and revised the following sentences in the Discussion as indicated below.
(Page 9, lines 329-330, in the Discussion)
We added “In summary, the present study revealed the differences in the weight of the SOFA sub-components.”
(Page 9, lines 332-338, in the Discussion)
We added “Further, our study clearly provided clinical evidence of organ cross-talk: in other words, the pathophysiology of multiple organ dysfunction is not simply one of additive combinations of each single organ dysfunction but one of a synergistically worsened condition with an extremely higher risk of death. It is therefore important when evaluating the severity of organ dysfunctions with the SOFA score to consider which organ and which combinations of organs are mainly responsible for the elevation of the total score.”
(Page 9, lines 338-340, in the Discussion)
Original sentence: “Our findings may thus be expected to be the foundation of future studies to construct an optimal method for clinically weighting the sub-components of SOFA.”
Revised sentence: “Our findings may also be expected to be the foundation of future studies to construct a new advanced method to evaluate the severity of organ dysfunction.”
- The title and abstract are misleading: the degree of deviation from the original SOFA score in my opinion warrants more recognition in these sections. One could argue that „biomarker components of the SOFA score” is a more accurate indication of this manuscript’s focus. The fact that patients with pre-existing organ dysfuntion could be treated as de-novo diagnosis of sepsis should be highlighted in the methods section.
Reply: Thank you for your valuable comments and suggestion. As you mentioned, a modified calculation method for SOFA components would cause bias from the true effect based on the original SOFA score. Accordingly, we revised the title and abstract as indicated below. Besides, possible elevation of the SOFA score associated with pre-existing organ dysfunction is one of the important limitations of the current study that might affect the patient selection and study results. We thus added the following sentences in the Materials and Methods to address this point more clearly.
Original title: “Non-linear and interaction analyses of SOFA subscores as predictive markers for sepsis: A nationwide retrospective study.”
Revised title: “Non-linear and Interaction Analyses of Biomarkers for Organ Dysfunctions as Predictive Markers for Sepsis: A Nationwide Retrospective Study”
(Page 1, lines 24-25, in the Abstract)
We added “The cardiovascular and neurological subscores were calculated by a modified method.”
(Page 1, lines 25-33, in the Abstract)
“Associations between the SOFA components and in-hospital mortality were examined using restricted cubic spline analyses. We also evaluated two-way interaction effects on mortality between an increase of one SOFA subscore and another. Restricted cubic spline analyses showed that an increase in total SOFA score was sharply and linearly associated with increased mortality. However, the prognostic association of subscores was varied widely by biomarker: platelet count showed a J-shaped association, creatinine showed an inverted J-shaped association, and bilirubin showed only a weak association with mortality. The joint odds ratios on mortality of two SOFA scores were synergistically and significantly increased compared to the sum of the single odds ratios, especially in cardiovascular-neurological, coagulation-hepatic, and renal-hepatic combinations.”
Revised sentences: “Associations between the biomarkers of the SOFA components and mortality were examined using restricted cubic spline analyses, which showed that an increase in the total modified SOFA score was linearly associated with increased mortality. However, the prognostic association of subscores varied widely: platelet count showed a J-shaped association, creatinine showed an inverted J-shaped association, and bilirubin showed only a weak association. We also evaluated interaction effects on mortality between an increase of one subscore and another. The joint odds ratios on mortality of two modified SOFA subscores were synergistically increased compared to the sum of the single odds ratios, especially in cardiovascular-neurological, coagulation-hepatic, and renal-hepatic combinations.”
(Page 1, line 33, in the Abstract)
Original wording: “total SOFA score”
Revised wording: “total modified SOFA score”
(Page 2, line 91 – Page 3, line 97, in the Materials and Methods)
Original sentence: “In this study, sepsis was defined as having a proven/suspected infection and the development of organ dysfunction (total SOFA score of ≥ 2 points at the time of admission), according to the Sepsis-3 criteria.”
Revised sentences: “In this study, sepsis was defined as having a proven/suspected infection and the development of organ dysfunction according to the Sepsis-3 criteria. Organ dysfunction in this study was defined as a total SOFA score of ≥2 points at the time of admission because pre-existing organ dysfunction was not sufficiently recorded and Sepsis-3 suggested that the baseline SOFA score could be assumed to be zero in patients not known to have preexisting organ dysfunction [1]”
- Did the authors follow any reporting guidelines? I could not find exact number of missing observations in the text or supplement. Stating that the amount of missingness was moderate is definitely not enough and details of the multiple imputation process need to be described - in the current form I would not know how to reproduce these findings and how successful the imputation actually was. Furthermore, I would argue that the MAR assumption is shaky regarding e.g. bilirubin - one could argue that bilirubin was likely to be normal in cases where the attending physician did not see indications to measure it.
Reply: Thank you for your valuable comments. There were 2548 (6.6%) missing data for creatinine, 3933 (10.1%) for bilirubin, and 2454 (6.3%) for platelet count. To address this information, we revised the following sentences. However, as you mentioned, the cause of missing data for the biomarkers might not be assumed as being “missing at random” in this database. Therefore, multiple imputation on the biomarkers might provide misleading results. Finally, we decided to delete the sensitivity analyses in this study and revised sentences as indicated below.
(Page 4, lines 163-166, in the Materials and Methods)
Original sentence: “Because there were moderate proportions of missing data, we performed sensitivity analyses using a multiple imputation technique to calculate missing values for biomarkers because the probability of missing data for these markers could be assumed not to depend on the unobserved data themselves (missing at random).”
Revised sentences: “As there were moderate proportions of missing data in the biomarkers (2548 [6.6%] for creatinine, 3933 [10.1%] for bilirubin, and 2454 [6.3%] for platelet count), missing values were not imputed in any of the regression models because the cause of the missing data for the biomarkers could not be assumed to be “missing at random”.”
(Page 4, lines 8-11, in the Materials and Methods in the original version)
We deleted: “We created 10 imputations for each missing value using the other available variables and then fit the desired models separately on each of the 10 imputed datasets and combined the results based on the concepts developed by Rubin.20”
(Page 8, lines 10-14, in the Results in the original version)
We deleted: “3.5. Result of multiple imputation
We performed non-linear regression analyses as sensitivity analyses by imputing the missing values using a multiple imputation technique. As a result, the non-linear associations between mortality and the biomarkers were similar to those without imputations (Figure S1).”
(References)
We deleted Reference 20; Rubin DB. Statistical matching using file concatenation with adjusted weights and multiple imputations. J Bus Econ Stat 4:87–94, 1986.
(Supplementary materials)
We deleted Figure S1 in the original version.
- Synergistic effect were analysed using a dichotomy (≥2 points in a particular subscore), while the manuscript overall seems to argue for a refined non-linear appraoch. How are these two approaches compatible?
Reply: Thank you for your valuable question. As you pointed out, the interaction effect of SOFA subscores on mortality was evaluated using dichotomy methods. However, they might be confused with non-linear cubic spline methods. To distinguish these two methods more clearly, we added the following sentences in the Materials and Methods as indicated below.
(Page 3, lines 136-138, in the Materials and Methods)
We added “The detailed association between the biomarkers of the SOFA subscores and mortality was evaluated using non-linear methods, and the interaction effects of modified SOFA subscores on mortality were evaluated by dichotomy methods.”
**Minor issues**
- The timeframe is extremely wide - this can be viewed as both a strength and limitation. As the definition of sepsis (and treatment guidelines) changed several times over the years, a sensitivity analysis looking more narrowly at specific periods of time could provide additional perspective on the heterogeneity of studied relationships.
Reply: Thank you for your valuable comment and suggestion. We fully agree with your comment that the long-term design in the present study was a possible source of bias that might influence the study results. In this study, all cases were retrospectively diagnosed as sepsis based on the Sepsis-3 criteria, having a proven/suspected infection and the development of organ dysfunction, and thus, a change in the definition of sepsis would not largely affect the study findings. However, time-related changes in the management of sepsis, and updates of the Surviving Sepsis guidelines, might affect the study results. Therefore, according to your suggestion, we classified the study population into two groups based on the time of hospital admission (the earlier group: February 1, 2006 to December 31, 2012 and the recent group: January 1, 2013 to December 31, 2019), and performed a subgroup analysis. As a result, the associations between the biomarkers and mortality were similar between the earlier and recent subgroups. To address the information, we added the following sentence and new Figure S1.
(Page 4, lines 167-171, in the Materials and Methods)
We added “To evaluate the possible differences of findings according to the time-related changes in the management of sepsis and updates of the Surviving Sepsis guidelines, we performed subgroup analysis based on the time of hospital admission (the earlier group: February 1, 2006 to December 31, 2012 and the recent group: January 1, 2013 to December 31, 2019).”
(Page 4, lines 185-189, in the Results)
We added “For the subgroup analysis, we classified the study population based on the time of hospital admission. Patients who were admitted to hospital from February 1, 2006 to December 31, 2012 were classified into the earlier group (n = 19,201), and patients who were admitted from January 1, 2013 to December 31, 2019 were classified into the recent group (n = 19,699).”
(Page 6, lines 215-217, in the Results)
We added “The results of restricted cubic spline analyses in the two subgroups based on the time of hospital admission are shown in Figure S1. The associations between the biomarkers and mortality were similar between the earlier and recent subgroups.”
(Supplementary materials)
We added new Figure S1 for the subgroup analyses.
- From my arguably limited understanding, Harrell recommends 5 knots in large datasets - what is the advantage of tweaking the knot positions based on statistical significance? I assume such supervised learning distorts the properties of p-values and confidence intervals. What were the final knot positions (percentile of data distribution)?
Reply: Thank you for your valuable comment and questions. Prof. Harrell recommends placing knots at equally spaced percentiles of the original variable’s marginal distribution. As you pointed out, five knots were determined as Harrell’s default percentiles, especially for a large-scale analysis. However, 5 knots is not always a suitable number in regard to the pathophysiological features of sepsis and each biomarker. We thus determined the number of knots by the Wald test. According to your suggestion, we describe the percentiles of the knots and thresholds of biomarkers in a new table (Table S3) as indicated below.
(Page 5, line 200, in the Materials and Methods)
We added “(Table S3)”
Table S3 Percentiles of knots and thresholds of biomarkers in the restricted cubic spline analyses. |
|||||
Biomarker |
Number of knots |
Threshold |
|||
Percentiles |
|||||
Platelet count (103/µL) |
4 |
6.1 |
13.4 |
19.2 |
34.1 |
5 |
35 |
65 |
95 |
||
Bilirubin (mg/dL) |
4 |
0.3 |
0.6 |
1.7 |
4.6 |
5 |
35 |
65 |
95 |
||
Creatinine (mg/dL) |
3 |
0.57 |
1.06 |
3.21 |
|
10 |
50 |
90 |
|||
SOFA total |
3 |
3 |
4 |
7 |
|
10 |
50 |
90 |
|||
SOFA: Sequential Organ Failure Assessment. |
- Please use the term multivariable not multivariate in this context (there is one dependent variable).
Reply: Thank you for your valuable suggestion. We revised this wording throughout the manuscript.
- The term confounding makes sense in a causal inference framework - is this the author’s intention? If so, STROBE guidelines and the exposure-outcome framework should be applied consistently. If not and prediction is of interest, the TRIPOD guidelines apply.
Reply: Thank you for your valuable suggestion. As you mentioned, the STROBE guidelines recommend that any exposures, confounders, or outcomes measurements be reported for the reader to critically appraise the study’s reliability and validity. We thus revised the manuscript as indicated below.
(Page 4, lines 155-161, in the Materials and Methods)
Original sentence: “Multivariate analysis included the five components of SOFA as covariates to adjust for possible confounding effects between the subscores.”
Revised sentences: “There were possible confounding effects on mortality between the SOFA subscores: in other words, patients having one organ dysfunction tended to have additional ones, which might enhance the risk of death. We therefore fitted multivariable analyses for SOFA subscore on mortality, including the other four SOFA subscores as covariates, to adjust for possible confounding effects and evaluate the independent risk of death associated with an increase of the subscore.”
- What treatment effects to the authors have in mind in the last paragraph of the methods section?
Reply: Thank you for noticing this. Actually, “treatment effects” is not correct. We are sorry about this and have revised the wording as indicated below.
(Page 4, line 173, in the Materials and Methods)
Original wording: “Because of the underpowered nature of the analyses investigating potential heterogeneities of treatment effects,…”
Revised wording: “Because of the underpowered nature of the interaction analysis,…”
- What does it mean that scores are „dependently associated” with an increase in mortality?
Reply: Thank you for noticing this. Actually, “independently” is correct. We are sorry about this and have revised the word.
- I would avoid the use of the word „skyrocketed” and ask the authors to look closely at confidence interval around the cardiovascular subscore of 4 - it’s just not enough data to draw conclusions about this group of patients.
Reply: Thank you for your valuable suggestion. Accordingly, we deleted the following sentences in the Results section.
(Page 6, line 15, in the Results in the original version)
We deleted: “, which skyrocketed when the subscore increased from 3 to 4 points”.
- „Despite the heterogenetic (and partly suboptimal) prognostic values” - did you mean heterogeneous? I would advice to use parentheses as sparsely as possible and rephrase the sentence.
Reply: Thank you for your valuable suggestion. Accordingly, we revised the following sentence in the Discussion section.
(Page 9, line 309, in the Discussion)
Original sentence: “Despite the heterogenetic (and partly suboptimal) prognostic values of the subscores”
Revised sentence: “Despite the heterogeneous and partly suboptimal prognostic values of the subscores”
- The authors argue that patients who are anesthesized and intubated automatically have higher scores in other systems. However, GCS is often assumed or carried forward using the last value before the patients received medications that alter mental status. Furthermore, SOFA uses the PaO2/FiO2 ratio, so intubation itself is less relevant than pulmonary function.
Reply: Thank you for your valuable comments. The reliability of the SOFA score in anesthetized and intubated patients is one of the methodological limitations of the current study. We thus added the following sentence in the Limitations.
(Page 10, lines 352-354, in the Discussion)
We added “Fourth, this study included anesthetized and intubated patients, who tend to have higher neurological and cardiovascular subscores compared to other subscores.”
- In the limitations the authors state that detailed amounts of catecholamines were not registered. How was the cardiovascular score calculated then?
Reply: Thank you for noticing this. Actually, detailed amounts of catecholamines were recorded in our datasets, and “vital signs” is correct at this location in the manuscript. We are sorry about this and have revised the wording as follows.
(Page 10, line 347, in the Discussion)
Original words: “detailed amounts of catecholamines administered”
Revised words: “vital signs”
**Other issues**
I had to re-read some sections, particularly methods - several sentences feel like they could have been made more readable but I will not make harsh judgements as I am not a native speaker myself. There are several typos in the text.
Reply: Thank you for your suggestion. We had our revised manuscript checked by an English language native speaker.

Reviewer 2 Report
The authors restropetively investigated a large cohort of patients with sepsis and showed the total SOFA score is good prognostic marker for mortality. This is not a new topic but with large number of patients, the data are reliable and will benefit clinical practice.
The weakness is the diagnosis crossing the periods of both sepsis-2 and sepsis-3 and may not be so accurate. The other is no specific analysis of underlying diseases that may affect mortality.
The minor point is the cutoff values and sensitivity and specificity shall be presented. In the figure 1, the 18,6125 shall be 186,125.
Author Response
We would like to thank all of the editors and reviewers for helping us improve our article. We revised our manuscript according to the comments and suggestions given by the reviewers. We also highlighted additional changes in the revised manuscript in red color with underline. Below are our point-by-point responses to the reviewers’ comments.
Reviewer #2:
The authors restropetively investigated a large cohort of patients with sepsis and showed the total SOFA score is good prognostic marker for mortality. This is not a new topic but with large number of patients, the data are reliable and will benefit clinical practice.
The weakness is the diagnosis crossing the periods of both sepsis-2 and sepsis-3 and may not be so accurate.
Reply: Thank you for your valuable comments. We fully agree with your comment that the long-term design in the present study was a possible source of bias that might influence the study results. In this study, all cases were retrospectively diagnosed as sepsis based on the Sepsis-3 criteria, having a proven/suspected infection and the development of organ dysfunction, and thus, a change in the definition of sepsis would not largely affect the study findings. However, time-related changes in the management of sepsis, and updates of the Surviving Sepsis guidelines, might affect the study results. Therefore, we classified the study population into two groups based on the time of hospital admission (the earlier group: February 1, 2006 to December 31, 2012 and the recent group: January 1, 2013 to December 31, 2019), and performed a subgroup analysis. As a result, the associations between the biomarkers and mortality were similar between the earlier and recent subgroups. To address the information, we added the following sentences and new Figure S1.
(Page 4, lines 167-171, in the Materials and Methods)
We added “To evaluate the possible differences of findings according to the time-related changes in the management of sepsis and updates of the Surviving Sepsis guidelines, we performed subgroup analysis based on the time of hospital admission (the earlier group: February 1, 2006 to December 31, 2012 and the recent group: January 1, 2013 to December 31, 2019).”
(Page 4, lines 185-189, in the Results)
We added “For the subgroup analysis, we classified the study population based on the time of hospital admission. Patients who were admitted to hospital from February 1, 2006 to December 31, 2012 were classified into the earlier group (n = 19,201), and patients who were admitted from January 1, 2013 to December 31, 2019 were classified into the recent group (n = 19,699).”
(Page 6, lines 215-217, in the Results)
We added “The results of restricted cubic spline analyses in the two subgroups based on the time of hospital admission are shown in Figure S1. The associations between the biomarkers and mortality were similar between the earlier and recent subgroups.”
(Supplementary materials)
We added new Figure S1 for the subgroup analyses.
The other is no specific analysis of underlying diseases that may affect mortality.
Reply: Thank you for your valuable comment. Actually, our dataset recorded the anatomical site of infection in septic patients. However, it is difficult to conduct subgroup analyses based on the site of infection because the reliability of the findings would decrease when the study population was subdivided into many subgroups. We thus decided to add this point as a study limitation in the Limitations as indicated below.
(Page 10, lines 354-355, in the Results)
We added “Fifth, the heterogenous pathophysiology of sepsis based on differences in the anatomical site of infection might have also affected the study findings .”
The minor point is the cutoff values and sensitivity and specificity shall be presented.
Reply: Thank you for your valuable comment. According to your suggestion, we conducted receiver operating characteristic (ROC) analyses and evaluated the area under the ROC curve, sensitivity, and specificity of SOFA score for mortality. To address the results, we added the following sentence.
(Page 4, lines 160-162, in the Materials and Methods)
We added “We also conducted receiver operating characteristic (ROC) analyses to evaluate the predictive value of the SOFA score for mortality.”
(Page 6, lines 204-206, in the Results)
We added “When the cutoff value was defined as 5 points, the sensitivity, specificity, and area under the ROC curve of the SOFA score for mortality were 38.1%, 86.3%, and 0.676, respectively.”
In the figure 1, the 18,6125 shall be 186,125.
Reply: Thank you for noticing this. We are sorry about this and corrected the comma placement.
Round 2
Reviewer 1 Report
Thank you for adressing most of my comments, I’m glad you’ve found them helpful and refined the manuscript accordingly. The paper has been improved significantly. Two comments have not been considered:
- Reporting guidelines. The paper seems to be on a crossroads between the STROBE and TRIPOD framework - I would like the authors to either attach one of these checklists or briefly justify not including either.
2. Interaction analyses - can the authors present continuous by continuous interactions, even if only in the supplement? (https://www.fharrell.com/post/addvalue/). I would suggest describing what exactly has been dichotomised instead of just using the umbrella term „dichotomy”.
Thank you for the opportunity to review the manuscript was again, I hope you will find these minor suggestions helpful.
Author Response
We would like to thank all of the editors and reviewers for helping us improve our article. We revised our manuscript according to the comments and suggestions given by the reviewers. We also highlighted additional changes in the revised manuscript in red color with underline. Below are our point-by-point responses to the reviewers’ comments.
Reviewer #1:
Thank you for adressing most of my comments, I’m glad you’ve found them helpful and refined the manuscript accordingly. The paper has been improved significantly. Two comments have not been considered:
- Reporting guidelines. The paper seems to be on a crossroads between the STROBE and TRIPOD framework - I would like the authors to either attach one of these checklists or briefly justify not including either.
Reply: Thank you for your valuable suggestion. Accordingly, we have attached the STROBE check list for the manuscript.
- Interaction analyses - can the authors present continuous by continuous interactions, even if only in the supplement? (https://www.fharrell.com/post/addvalue/). I would suggest describing what exactly has been dichotomised instead of just using the umbrella term „dichotomy”.
Reply: Thank you for your kind and valuable suggestions. Regarding the first suggestion, unfortunately, it is technically difficult for us to accurately capture the proportion of the joint effect attributable to additive interaction in continuous variables. For the second suggestion, we revised the manuscript as indicated below.
(Page 3, lines 138-139, in the Materials and Methods)
Original sentence: “by dichotomy methods.”
Revised sentence: “by dichotomizing each SOFA subscore into the higher score (2-point or more) or the lower score (1-point or less).”
Thank you for the opportunity to review the manuscript was again, I hope you will find these minor suggestions helpful.